# Nitrogen-Rich Triazine-Based Covalent Organic Frameworks as Efficient Visible Light Photocatalysts for Hydrogen Peroxide Production

**DOI:** 10.3390/nano14070643

**Published:** 2024-04-08

**Authors:** Shu Yang, Keke Zhi, Zhimin Zhang, Rukiya Kerem, Qiong Hong, Lei Zhao, Wenbo Wu, Lulu Wang, Duozhi Wang

**Affiliations:** 1College of Chemistry, Xinjiang University, Urumqi 830017, China; ys18536398768@sina.com (S.Y.); z18190914@sina.com (Z.Z.); ruky_k223@sina.com (R.K.); hongqiong0809@sina.com (Q.H.); 14719415415m@sina.cn (L.Z.); ww100605249@sina.com (W.W.); 2State Key Laboratory of Chemistry and Utilization of Carbon Based Energy Resources, Urumqi 830017, China; 3College of Engineering, China University of Petroleum—Beijing at Karamay, Karamay 834000, China; zhikeke@cupk.edu.cn; 4State Key Laboratory of Heavy Oil Processing—Karamay Branch, Karamay 834000, China

**Keywords:** imine linked, covalent organic framework, hydrogen peroxide, photocatalysis, two-electron process

## Abstract

Covalent organic frameworks (COFs) have been widely used in photocatalytic hydrogen peroxide (H_2_O_2_) production due to their favorable band structure and excellent light absorption. Due to the rapid recombination rate of charge carriers, however, their applications are mainly restricted. This study presents the design and development of two highly conjugated triazine-based COFs (TBP-COF and TTP-COF) and evaluates their photocatalytic H_2_O_2_ production performance. The nitrogen-rich structures and high degrees of conjugation of TBP-COF and TTP-COF facilitate improved light absorption, promote O_2_ adsorption, enhance their redox power, and enable the efficient separation and transfer of photogenerated charge carriers. There is thus an increase in the photocatalytic activity for the production of H_2_O_2_. When exposed to 10 W LED visible light irradiation at a wavelength of 420 nm, the pyridine-based TTP-COF produced 4244 μmol h^−1^ g^−1^ of H_2_O_2_ from pure water in the absence of a sacrificial agent. Compared to TBP-COF (1882 μmol h^−1^ g^−1^), which has a similar structure but lacks pyridine sites, TTP-COF demonstrated nearly 2.5 times greater efficiency. Furthermore, it exhibited superior performance compared to most previously published nonmetal COF-based photocatalysts.

## 1. Introduction

In recent years, environmental and energy issues have received increasing attention, which has promoted the rapid development of clean energy. Hydrogen peroxide (H_2_O_2_) is considered a viable sustainable energy source due to its efficiency, versatility, and environmentally friendly oxidant. Therefore, H_2_O_2_ is widely used in several industries, including chemical synthesis, fuel cells, wastewater treatment, and sterilization [1]. H_2_O_2_ is traditionally produced on a large scale using anthraquinone oxidation, a process that requires the use of high temperatures and pressures. Moreover, anthraquinone oxidation consumes a significant quantity of costly hydrogen gas and results in the production of hazardous organic by-products [2]. An alternative technique for generating H_2_O_2_ involves the direct chemical synthesis of H_2_ and O_2_. However, the O_2_ and H_2_ gases must be thoroughly diluted with inert gas to avoid an explosion [3]. Due to these factors, a safe, efficient, and clean H_2_O_2_ generation system is urgently required. Photocatalysis shows excellent potential for the production of H_2_O_2_. Overall, there are two strategies to achieve H_2_O_2_ photosynthesis: one involves a water oxidation reaction (WOR), and the other proceeds through the oxygen reduction reaction (ORR) [4]. H_2_O_2_ can be produced via two photocatalytic WOR routes: one-step direct 2e^−^ processes (Equation (1)) and two-step 2e^−^ processes (Equations (2) and (3)). The photocatalytic ORR generates H_2_O_2_ through two distinct processes: a one-step direct 2e^−^ reaction (Equation (4)) and a two-step process comprising two 2e^−^ reactions (Equations (5) and (6)) [5].
(1)$$2H_{2}O+2h^{+}\to H_{2}O_{2}+2H^{+}$$
(2)$$H_{2}O+h^{+}\to H^{+}+\cdot \;OH$$
(3)$$\cdot \;OH+\cdot \;OH\to H_{2}O_{2}$$
(4)$$O_{2}+2H^{+}+2e^{-}\to H_{2}O_{2}$$
(5)$$O_{2}+e^{-}\to \cdot \;O_{2}^{-}$$
(6)$${\cdot \;O}_{2}^{-}+2H^{+}+e^{-}\to H_{2}O_{2}$$

Currently, the most often used photocatalysts for the production of H_2_O_2_ are TiO_2_ [6], g-C_3_N_4_ [7], and some semiconductors [8]. These catalysts are used due to their beneficial characteristics, such as excellent chemical and thermal stability, cost-effectiveness, minimal toxicity, and corrosion resistance. However, their application in the process of H_2_O_2_ photosynthesis continues to encounter two primary obstacles: (i) The band gaps of these catalysts are relatively broad [9]; (ii) photoinduced e^−^ and h^+^ are rapidly recombined and cannot migrate to their surface to participate in the reaction [10]. Hence, it is crucial to develop highly effective photocatalysts with appropriate photo-redox ability. A characteristic class of reticular materials is known as covalent organic frameworks (COFs), where the molecular building blocks are joined through covalent bonds to form extended structures that can be either two-dimensional (2D) [11] or three-dimensional (3D) [12]. COFs have garnered considerable attention for their well-organized crystalline structure, adaptable topology, and numerous redox-active sites. This has led to significant interest in their application in various fields, including gas storage [13], adsorption [14], energy storage [15], photocatalytic hydrogen production [16], and CO_2_ reduction [17]. Specifically, 2D triazine-containing COFs, commonly referred to as covalent triazine frameworks (CTFs), exhibit significant promise in the field of photocatalysis. CTFs have a high nitrogen content, which makes them highly sensitive to visible light. They demonstrate high adjustability, completely conjugated structures, and excellent chemical stability [18]. Xu and colleagues have proven that the presence of acetylene (-C≡C-) or diacetylene (-C≡C-C≡C-) moieties can greatly enhance the photocatalytic production of H_2_O_2_ in CTFs [19]. However, the majority of reported CTFs exhibit minimal crystallinity or exist in an amorphous state. This greatly affects the separation of charge carriers and the process of photoexcitation, leading to a decrease in the photocatalytic activity of the carriers [20]. To achieve a highly crystalline structure of COFs, a commonly used method is to directly couple triazine derivatives to other monomers by a Schiff base condensation process. This leads to the formation of an imine-linked 2D COF with a triazine core [21]. For instance, a triazine-structured imine-linked COF was reported by Han et al. Due to its improved crystallinity, broader visible light absorption range, and enhanced charge separation and transfer characteristics, this structure facilitates the efficient photocatalytic generation of H_2_O_2_ [22]. The catalytic efficiency of COFs for H_2_O_2_ generation can be greatly improved by rationally adding appropriate sites, such as cyano groups, thioether-decorated sulfone units, piperazine sites, and others. However, COFs continue to require improvements in their photoactivity as a result of restrictions on electron transport within their frameworks, rapid recombination of photogenerated charges, and inefficient generation of intrinsic charges [23]. Therefore, to facilitate electron migration and reduce electron–hole recombination, it is crucial to develop effective strategies for controlling charge separation. Heteroatom doping is an effective method for controlling the optical and electronic characteristics of nanostructures at the atomic scale. The incorporation of distinct heteroatoms (e.g., N, O, S) into the skeleton of COFs is achieved through the precise design of building blocks [24]. The presence of heteroatoms in the skeleton of COFs inevitably results in a modification of their energy bands, thereby causing effective sites to be introduced and alterations to their electronic and optical properties [25]. Nitrogen doping exhibits notable advantages in various domains when compared to other heteroatoms. Firstly, N and C are adjacent elements in the periodic table and have comparable atomic radii, which prevents the issue of lattice mismatch [26]. Secondly, incorporating N into the structure of COFs not only enhances their inherent physicochemical properties but also significantly improves their surface wettability, electron-donation, electrical conductivity, and reactivity [27]. The presence of a lone pair of electrons on the nitrogen atom in pyridine results in an electron-rich state, developing a partial negative charge on the nitrogen atom. This, in turn, induces a positive charge on the surrounding carbon atoms, leading to a p-type effect. This unique electronic structure contributes to the important role of pyridine nitrogen in different applications. In oxygen reduction catalytic reactions, the pyridine nitrogen is considered to be the active center of the catalytic reaction.

The addition of nitrogen-containing functional groups to a COF skeleton can provide several active sites and significantly enhance photocatalytic performance. In this work, a nitrogen-rich triazine structure was used to prepare an imine-type COF with a high conjugation degree, denoted TBP-COF. To enhance the nitrogen content, an imine-type TTP-COF was synthesized by combining a triazine structure with a pyridine ring. These two COFs demonstrate significantly enhanced abilities to absorb light and have modified energy band structures, resulting in improved production of photocatalytic H_2_O_2_. TTP-COF is more hydrophilic and nucleophilic due to the nitrogen-rich skeleton structure provided by its triazine and pyridine structures. As a result, TTP-COF produces 4244 μmol h^−1^ g^−1^ of H_2_O_2_ from water in the absence of a sacrificial agent when exposed to 10 W LED visible light irradiation (λ = 420 nm). The efficiency of TTP-COF is approximately 2.5 times more than that of TBP-COF (1882 μmol h^−1^ g^−1^), although it has a comparable structure without pyridine sites. This study provides an in-depth comprehension of the systematic development of highly effective catalysts for the photocatalytic synthesis of H_2_O_2_ from O_2_ and water without the need for sacrificial agents_._ This study illustrates the essential functions of surface hydrophilicity, O_2_ adsorption behavior, and the separation of e^−^–h^+^ pairs in controlling the 2e^−^ ORR process.

## 2. Materials and Methods

### 2.1. Materials

The materials and reagents involved in this work are described in the Appendix A.

### 2.2. Preparation of TBP-COF and TTP-COF

#### 2.2.1. Synthesis of TBP-COF

2,4,6-tris(4-bromophenyl)-1,3,5-triazine (TBP-Br): As shown in Appendix A, in a round bottom flask, 3.54 mL (6 g, 40 mmol) of trifluoromethanesulfonic acid was slowly added to 2 g (11 mmol) of 4-Bromobenzonitrile at 0 °C and stirred for 30 min. The mixture was then stirred at room temperature overnight. After completion of the reaction, the resultant mixture was washed with 100 mL of deionized water, yielding a large amount of white solid precipitate, which was then filtered and collected. The product was purified to yield 1.95 g of a white solid (yield, 99%) [28].4′,4‴,4′′′′′-(1,3,5-triazine-2,4,6-triyl) tris (([1,1′-biphenyl]-4-amine)) (TBP-NH_2_): As shown in Appendix A, TBP-Br (0.546 g, 1.0 mmol), 4-aminophenylboronic acid pinacol ester (0.788 g, 3.6 mmol), Pd(PPh_3_)_4_ (119.0 mg, 0.1 mmol), and K_2_CO_3_ (0.513 g, 3.72 mmol) were added to a two-neck flask and subjected to a vacuum for 15 min. The mixture was then heated at 110 °C for 14 h with the addition of dioxane (37 mL) and H_2_O (5 mL). The solution was poured into a stirring beaker containing ice cubes and H_2_O. The solid was separated through suction filtration, placed in a beaker, and treated with a small amount of methanol (MeOH). Following suction filtration, the undissolved solid was oven-dried for 12 h, yielding 0.472 g of green solid (yield, 81%).^1^H NMR (Appendix A) (600 MHz, DMSO-d_6_) δ = 8.73 (d, *J* = 8.6 Hz, 6H), 7.85 (d, *J* = 8.6 Hz, 6H), 7.56 (d, *J* = 8.7 Hz, 6H), 6.71 (d, *J* = 8.5 Hz, 6H), 5.44 (s, 6H) (Appendix A).4′,4‴,4′′′′′–(1,3,5-triazine–2,4,6-triyl) tris (([1,1′-biphenyl]-4-carbaldehyde)) (TBP-CHO): As shown in Appendix A, TBP-Br (0.546 g, 1.0 mmol), 4-formylphenylboronic acid (0.788 g, 3.6 mmol), Pd(PPh_3_)_4_ (119.0 mg, 0.1 mmol), and K_2_CO_3_ (0.513 g, 3.72 mmol) were added in a two-neck flask and placed under vacuum for 15 min. Following the addition of dioxane (37 mL) and H_2_O (5 mL), the mixture was heated for 14 h at 110 °C. The mixture was transferred into a stirring beaker containing ice cubes and H_2_O. The solid was separated by suction filtration, placed in a beaker, and treated with a small amount of MeOH; the mixture was heated until it boiled, and then sonicated for 15 min. After suction filtration, the undissolved solid was treated with acetone and dried in an oven for 12 h, yielding 0.485 g of gray solid (yield, 78%) [29].^1^H NMR (Appendix A) (600 MHz, TFA) δ = 9.98 (s, 3H), 8.82 (d, *J* = 8.5 Hz, 6H), 8.15 (d, *J* = 8.3 Hz, 6H), 8.05 (d, *J* = 8.5 Hz, 6H), 7.96 (d, *J* = 8.3 Hz, 6H) (Appendix A).TBP-COF: In Figure 1, TBP-CHO (31 mg, 0.05 mmol) and TBP-NH_2_ (29 mg, 0.05 mmol) were mixed in 35 mL Pyrex tubes. The solvents *o*-dichlorobenzene (*o*-DCB) and *n*-butanol were added in a 1:1 volume ratio (4 mL). Additionally, acetic acid (6 M, 0.50 mL) was included. The mixture was subjected to ultrasonication for 15 min to achieve uniform dispersion, sealed in a pressure-resistant glass tube, and reacted in an oil bath at 120 °C for 5 d. After the reaction was completed, the reaction tube was cooled to room temperature, and then the crude product was collected and washed three times with N,N-Dimethylformamide (DMF), acetone, and tetrahydrofuran (THF). After that, it was dried for 24 h at 100 °C to obtain the green TBP-COF compound (41 mg), with a 68% yield.

#### 2.2.2. Synthesis of TTP-COF

2,4,6-tris(5-bromopyridin-2-yl)-1,3,5-triazine (TTP-Br): As shown in Appendix A, NH_4_Br (65 mg) and diisopropylethylamine (113 μL, 0.66 mmol) were added into a round-bottom flask containing a suspension of 5-Bromopicolinonitrile (1.0 g) in 1-pentanol (5 mL). The suspension was heated while being stirred for 12 h in an oil bath at 135 °C. After the reaction, the mixture was cooled to room temperature, resulting in a significant amount of solid precipitation in the flask. The yellow–white solid (0.8 g) was obtained (yield, 82%) after the solid was collected and filtered, then thoroughly washed with acetonitrile [30].4,4′,4″-((1,3,5-triazine-2,4,6-triyl) tris (pyridine-6,3-diyl)) trianiline (TTP-NH_2_): As shown in Appendix A, TTP-Br (0.549 g, 1.0 mmol), 4-aminophenylboronic acid pinacol ester (0.788 g, 3.6 mmol), Pd(PPh_3_)_4_ (119.0 mg, 0.1 mmol), CsF (0.547 g, 3.6 mmol), and Cs_2_CO_3_ (1.212 g, 3.72 mmol) were added to a two-neck flask and kept under vacuum for 15 min. The mixture was then heated at 110 °C for 14 h after adding dioxane (37 mL) and H_2_O (5 mL). The solution was poured into a stirring beaker containing ice cubes and H_2_O. The solid was separated by suction filtration, transferred to a beaker, and treated with a small amount of MeOH. After suction filtration, the undissolved solid was oven-dried for 12 h, yielding 0.422 g of orange solid (yield, 72%).^1^H NMR (Appendix A) (600 MHz, DMSO-d_6_) δ = 9.15 (s, 3H), 8.76 (d, *J* = 8.3 Hz, 3H), 8.26 (d, *J* = 5.9 Hz, 3H), 7.63 (d, *J* = 8.5 Hz, 6H), 6.74 (d, *J* = 8.5 Hz, 6H), 5.56 (s, 6H) (Appendix A).^13^C NMR (Appendix A) (151 MHz, DMSO-d_6_) δ = 171.78, 150.47, 150.27, 147.34, 139.05, 133.17, 128.36, 125.42, 123.16, 114.80, 39.58 (Appendix A).HRMS *m*/*z* [M^+^] calculated for C_36_H_28_N_9_: 586.24622; found: 586.24648.4,4′,4″–((1,3,5–triazine–2,4,6-triyl) tris (pyridine–6,3–diyl)) tribenzaldehyde (TTP-CHO): As shown in the Appendix A, TTP- Br (0.549 g, 1.0 mmol), 4-formylphenylboronic acid (0.788 g, 3.6 mmol), Pd(PPh_3_)_4_ (119.0 mg, 0.1 mmol), CsF (0.547 g, 3.6 mmol), and Cs_2_CO_3_ (1.212 g, 3.72 mmol) were added in a two-neck flask and subjected to a vacuum for 15 min. Afterwards, 37 mL of dioxane and 5 mL of H_2_O were added, and the mixture was heated to 110 °C for 14 h. The solution was poured into a stirring beaker containing ice cubes and H_2_O. The solid was separated by suction filtration, placed in a beaker, and treated with a small amount of MeOH; the mixture was heated till boiling, and then sonicated for 15 min. Following suction filtration, the undissolved solid was dried in an oven for 12 h, yielding 0.487 g of yellow solid (yield, 78%).^1^H NMR (Appendix A) (600 MHz, TFA) δ = 10.11 (s, 3H), 9.74 (d, *J* = 2.2 Hz, 3H), 9.66 (d, *J* = 8.4 Hz, 3H), 9.28 (dd, *J* = 8.4, 2.2 Hz, 3H), 8.32 (d, *J* = 8.4 Hz, 6H), 8.12 (d, *J* = 8.4 Hz, 6H) (Appendix A).^13^C NMR (Appendix A) (151 MHz, TFA) δ = 192.24, 162.12, 141.98, 140.87, 138.77, 137.31, 134.33, 132.96, 127.66, 124.55, 124.12 (Appendix A).HRMS *m*/*z* [M^+^] calculated for C_39_H_25_O_3_N_6_: 625.19827; found: 625.19880.TTP-COF: As shown in Figure 2, TTP-CHO (31 mg, 0.05 mmol) and TTP-NH_2_ (29 mg, 0.05 mmol) were introduced into 35 mL Pyrex tubes. Subsequently, a mixture of mesitylene:1, 4-dioxane solvents (4 mL, *v*:*v* = 1:1), and acetic acid (6 M, 0.5 mL) was added. After 15 min of ultrasonication to ensure uniform dispersion, the mixture was sealed in a pressure-resistant glass tube and reacted for 5 d at 120 °C in an oil bath. After the reaction was completed, the reaction tube was cooled to room temperature, and then the crude product was collected and washed three times with DMF, acetone, and THF. It was then dried at 100 °C for 24 h to give the croci TTP-COF compound (49 mg), with a yield of 82%.

### 2.3. Characterization

The ^1^H NMR spectra were recorded using a 600 MHz spectrometer (Bruker, Salbrücken, Germany). Solid-state ^13^C NMR spectra were collected using a 400 MHz NMR spectrometer (Bruker, Salbrücken, Germany). An X-ray photoelectron spectrometer (Thermo Scientific K-Alpha, MA, USA) was used to conduct X-ray photoelectron spectroscopy (XPS). The FT-IR spectra were obtained using a Brucker Equinox 55FT-IR spectrometer from Salbrücken, Germany. The spectra were collected in the range of 4000 to 400 cm^−1^. Powder X-ray diffraction (XRD) patterns were obtained using the Bruker D8 (Salbrücken, Germany). The UV-vis spectrophotometer (U3900, Kyoto, Japan) was used to acquire optical diffuse reflectance spectra at room temperature. A white standard of BaSO_4_ was used as the reference. Transmission electron microscopy (TEM) images were obtained using a SU8200 (Kyoto, Japan). An FL spectrofluorometer (F-4500, Kyoto, Japan) was used to collect steady-state photoluminescence (PL) spectra. Meanwhile, time-correlated single photon counting measurements were performed to acquire photoluminescence decay profiles using a single photon counting controller (FLS1000, Edinburgh, UK). The catalyst was subjected to an oxygen temperature-programmed desorption (O_2_–TPD) test using an AutoChem1 II 2920 instrument from Norcross, GA, USA. The free radicals of O_2_^·−^ were recorded using electron paramagnetic resonance (EMX nano, Bruker, Salbrücken, Germany).

### 2.4. Theoretical Calculation Details

The Vienna Ab initio Simulation Package (VASP) was used to perform all density functional theory (DFT) calculations in the generalized gradient approximation (GGA) with the Perdew–Burke–Ernzerhof (PBE) functional.

The Gibbs free energy (ΔG) for each electrochemical process is calculated as
ΔG = ΔE + ΔE_ZPE_ − TΔS = D + D − D
where the values of ΔE, ΔE_ZPE,_ and ΔS represent the changes in DFT energy, zero-point energy, and entropy at a specific temperature of 298.15 K, respectively.

### 2.5. H_2_O_2_ Detection Methods

A septum was used to seal the tube after 2.5 mg of photocatalyst and 10 mL of water were added. After that, the suspension underwent ultrasonication treatment to disperse the photocatalyst. The suspension was then allowed to degas for 15 min by bubbling O_2_, and to attain absorption–desorption equilibrium, the mixed solution was stirred for 15 min in the dark. The reaction mixture was then illuminated using a 10 W, 420 nm LED. During the experiment, the temperature of the reaction system was carefully controlled at 25 °C using cooling water.

### 2.6. AQY Measurements

The AQY was calculated as follows:$$AQY=\frac{\left(number\;of\;H_{2}O_{2}\;production\right)\times 2}{number\;of\;incident\;photons}\times 100\%$$

The number of incident photons is
$$N_{incident}=\frac{Pt}{h\vartheta }=\frac{Pt\lambda }{hc}=\frac{ISt\lambda }{hc}$$
where I represent the light power intensity (W cm^−2^), S is the irradiation area (cm^2^), t is the reaction time (s), λ is the wavelength length (m) of the monochromatic light, h is the Planck’s constant (6.63 × 10^−34^ J S^−1^), and c is the speed of light in free space (3.0 × 10^8^ m s^−1^).

## 3. Results

### 3.1. Structural Analysis of TBP-COF and TTP-COF 

According to the Fourier transform infrared (FT-IR) spectra of TTP-COF and TBP-COF, the stretching peaks associated with -NH_2_ (3100–3400 cm^−1^) and C=O (1700 cm^−1^) of the precursors disappeared after synthesis. On the contrary, new peaks (1627 and 1609 cm^−1^) attributed to the stretching vibration of the C=N bond were observed in both COFs. It was demonstrated that the polymerization processes the triazine structure; no changes have occurred in TBP-COF and TTP-COF due to the presence of telescopic vibrational peaks at 1370 and 1352 cm^−1^, respectively, corresponding to the triazine structure (Figure 1a). This finding supports the successful Schiff base condensation reaction. Powder XRD patterns showed that both COFs are amorphous. Only diffraction peaks corresponding to the partially stacked structures, resulting from the strong Π–Π interaction between highly conjugated polymer layers, were found (Figure 1b). The solid-state ^13^C nuclear magnetic resonance (^13^C NMR) spectra [31] of TBP-COF and TTP-COF displayed signals at 158 and 162 ppm, corresponding to the carbon atoms of the imine bond (Figure 1c). 

The surface area and pore volume of TBP-COF and TTP-COF were analyzed by measuring the nitrogen adsorption–desorption isotherm at 77 K using the Brunauer–Emmett–Teller (BET) method. The BET-specific surface area was calculated using the BET model, resulting in specific surface areas of 73.58 m^2^ g^−1^ for TBP-COF and 22.42 m^2^ g^−1^ for TTP-COF (Appendix A), respectively. Furthermore, TBP-COF and TTP-COF had pore diameters of approximately 17.58 nm and 31.13 nm (Appendix A), as shown by the calculated pore size distributions.

Characteristic peaks of the C, N, and O elements were observed in the elemental composition spectra of TBP-COF (Figure 2a) and TTP-COF (Figure 2d), as determined by XPS. The presence of oxygen and water adsorbed on TBP-COF and TTP-COF could explain the observation of oxygen. The valence states of TBP-COF (Figure 2b,c) and TTP-COF (Figure 2e,f) were analyzed using XPS [32]. This confirmed the successful formation of the desired chemical structures in both COFs.

The morphology of TBP-COF and TTP-COF was analyzed using scanning electron microscopy (SEM) and transmission electron microscopy (TEM). As illustrated in Figure 3a, TBP-COF displayed a spherical morphology, whereas TTP-COF exhibited a nanosheet-like structure (Figure 3e). The TEM result further confirms that TBP-COF (Figure 3b) and TTP-COF (Figure 3f) are composed of numerous uniform nanostrips. Energy-dispersive X-ray spectroscopy (EDS) was used to confirm the uniform distribution of C and N elements in TBP-COF (Figure 3c,d) and TTP-COF (Figure 3g,h). The thermal stability of TBP-COF and TTP-COF was assessed using thermogravimetric (TG) analysis in a nitrogen atmosphere. TTP-COF showed slight mass loss between 30 °C and 100 °C (Appendix A), which might be related to the decomposition of COF oligomers and the evaporation of a small amount of water [33]. The thermal decomposition of TBP-COF and TTP-COF occurred at temperatures of 550 °C and 500 °C, respectively. In addition, both COFs showed excellent chemical stability in commonly used solvents (Appendix A). The C, H, and N contents of TBP-COF (Table 1) and TTP-COF (Table 2) were found to be in good agreement with the theoretical values, as confirmed by elemental analysis (EA).

### 3.2. Photocatalytic H_2_O_2_ Production Performance

As illustrated in Figure 4a, the UV–visible diffuse reflection spectra (UV-vis DRS) of both COFs revealed strong adsorption in the visible spectrum. The band gaps of TTP-COF and TBP-COF were both calculated using Tauc plots and found to be 2.05 eV and 2.50 eV, respectively (Figure 4b). TTP-COF exhibits a lower band gap and a broader range of visible light absorption due to the presence of a pyridine ring, in contrast to TBP-COF. This characteristic is advantageous for the photocatalytic production of H_2_O_2_. The electrochemical Mott–Schottky plots of both COFs (Figure 4c) displayed positive slopes, suggesting that they possess the characteristics of n-type semiconductors [34]. TBP-COF showed a flat band potential of −0.35 V (vs. Ag/AgCl), while TTP-COF had a flat band potential of −0.50 V (vs. Ag/AgCl). The valence band (VB) and conduction band (CB) were determined using the band gap and flat band calculations (Figure 4d). The results confirm that COFs have the ability to facilitate the 2e-ORR pathway, thereby ensuring a significant increase in H_2_O_2_ production (Figure 4e).

In the absence of any sacrificial reagent, the photocatalytic activity of the COFs in producing H_2_O_2_ was evaluated in water and O_2_ under 10 W LED visible light irradiation. Figure 5a illustrates that when exposed to 10 W LED visible light irradiation (λ = 420 nm), TBP-COF produced a moderate amount of H_2_O_2_ (1882 μmol h^−1^ g^−1^). The pyridine-containing structure TTP-COF increased the rate of H_2_O_2_ production by 4244 μmol h^−1^ g^−1^. Therefore, pyridine is essential for enhancing the activity of COFs towards H_2_O_2_. Figure 5b,c and Appendix A demonstrate that TTP-COF had superior performance compared to most other metal-free photocatalysts under LED visible light conditions in pure water for the production of H_2_O_2_. Furthermore, it outperformed most catalysts with sacrificial agents [19,23,35,36,37,38]. Thus, TTP-COF is identified as one of the most effective COF-based catalysts for photocatalytic H_2_O_2_ generation. To further examine the performance of TTP-COF in the production of H_2_O_2_, the dosage of the catalyst was examined. It can be observed from Figure 5d that the highest H_2_O_2_ production occurs at an optimal TTP-COF concentration of 0.25 g L^−1^; however, as the TTP-COF concentration is increased, the H_2_O_2_ concentration decreases. This is primarily attributed to the hindrance of light absorption and utilization caused by an excessively high catalyst concentration [39]. The photocatalytic pathway of H_2_O_2_ production over the COFs was elucidated through a series of contrast experiments. The photocatalytic performance of the COFs studied in the dark and under different atmospheric conditions (N_2_/Air/O_2_) is displayed in Figure 5e. H_2_O_2_ is produced via photocatalytic oxygen reduction, as almost no H_2_O_2_ was detected in an N_2_-saturated solution or under dark conditions [40]. In addition, quenching experiments were performed to gain insight into the role of reactive species, as illustrated in Figure 5f. KBrO_3_, MeOH, and p-benzoquinone (p-BQ) were employed as scavengers for e^−^, h^+^, and ·O_2_^−^, respectively [41]. The findings indicated that the H_2_O_2_ production of TTP-COF and TBP-COF significantly decreased when electrons and O_2_^·−^ were quenched, indicating that the ORR plays a crucial role in this photocatalytic system [42].

H_2_O_2_ is known to have poor stability, especially under light irradiation. Therefore, the decomposition of H_2_O_2_ on the developed catalysts was also investigated. The lack of significant variation in the concentration of the H_2_O_2_ aqueous solution when both COF catalysts are present suggests that the distinct H_2_O_2_ generation characteristics of the catalysts cannot be ascribed to differences in H_2_O_2_ decomposition (Figure 6a). At 420 nm, TBP-COF had an apparent quantum yield of 1% (Figure 6b), while the apparent quantum yield of TTP-COF was 5.2% (Figure 6c). After undergoing four rounds of irradiation, the recycling test results demonstrated that both TBP-COF and TTP-COF maintained a significantly high level of photocatalytic performance (Figure 6d). Furthermore, the FT-IR spectra showed no noticeable changes after the reaction. (Figure 6e,f). The characteristic peaks of C 1s, N 1s, and O 1s in TBP-COF and TTP-COF, as shown in Appendix A, did not show any noticeable peak shifts during the photocatalytic reaction. This provides further evidence of the structural integrity of the frameworks without decomposition. The morphology of TBP-COF and TTP-COF following photocatalysis was analyzed using SEM and TEM. TBP-COF retained its spherical form, but TTP-COF exhibited a nanosheet structure (Appendix A). EDS measurements revealed a homogenous distribution of C and N elements. However, the sizes of TBP-COF and TTP-COF exhibited increased surface roughness, thereby facilitating the scattering of incident light and enhancing light absorption capabilities [43]. These findings indicate that TBP-COF and TTP-COF demonstrate excellent stability in continuous H_2_O_2_ production, making them highly promising for real-time applications.

Dimethyl-1-pyrroline N-oxide (DMPO) was used as a spin-trap agent to obtain electron paramagnetic resonance (EPR) spectra. This facilitated the determination of the active oxygen species participating in the photocatalytic process [44]. The spectra of TTP-COF and TBP-COF displayed distinct peaks associated with DMPO-O_2_^·−^ when exposed to visible light (Figure 7a). This suggests that O_2_^·−^ was generated as an intermediate species. Furthermore, this result suggests that H_2_O_2_ is produced by this photocatalytic system via a 2e^−^ two-step ORR mechanism (O_2_ → O_2_^·−^ → H_2_O_2_). The O_2_ and H_2_O uptake performance of TBP-COF and TTP-COF was analyzed to understand the mechanism behind the enhanced catalytic performance of TBP-COF and TTP-COF for H_2_O_2_ production. The obtained O_2_ temperature-programmed desorption (O_2_-TPD) curves demonstrated that TTP-COF shows a more pronounced O_2_ desorption peak in comparison to TBP-COF (Figure 7b). Therefore, the pyridine-containing TTP-COF exhibits an enhanced chemical adsorption capacity for O_2_ [45]. In addition, because of the hydrophilic and nucleophilic properties of pyridine, the zeta potential (Figure 7c) and hydrophilicity (Figure 7d,e) of TTP-COF were significantly greater than those of TBP-COF. Furthermore, the hydrophilic and nucleophilic characteristics of pyridine [38] contribute to the increased chemisorption of O_2_ and enhanced hydrophilicity of TTP-COF, hence promoting the generation of H_2_O_2_ during the ORR process. To acquire more comprehensive insights into the charge transport mechanisms and visible light reactions of TBP-COF and TTP-COF, the electrochemical impedance spectra (EIS) and transient photocurrent responses of the photocatalysts were measured under light illumination [46]. The Nyquist curve of TTP-COF demonstrated a reduced diameter of the semicircle in comparison to TBP-COF (Figure 7f). This suggests that the pyridine group modification efficiently accelerates the kinetics of interfacial charge transport [47]. As demonstrated in Figure 7g, the introduction of pyridine increased the generation of photoexcited charges and facilitated the separation of photogenerated e^−^–h^+^ as a result of the lower photocurrent density of TBP-COF in comparison to TTP-COF. Time-resolved photoluminescence (TRPL) and photoluminescence (PL) spectroscopy were employed to analyze the behavior of photogenerated carriers in the two COFs. The PL intensity of TTP-COF was somewhat quenched in contrast to TBP-COF (Figure 7h), suggesting suppressed recombination of photogenerated electrons and holes [48]. Compared to TTP-COF (τAve. = 1.54 ns), TBP-COF exhibited a significantly longer average lifetime (τAve. = 4.20 ns), as illustrated in Figure 7i. The reduced lifetime suggests improved separation of electrons and holes, indicating that the photogenerated carriers are effectively captured by reactive substrates, enabling them to facilitate redox reactions [49]. The findings of this study indicate that the pyridine group modification accelerates interfacial charge transport kinetics and facilitates the separation of photogenerated carriers. As a result, TTP-COF demonstrates improved photocatalytic efficacy in the production of H_2_O_2_.

Quantum chemical modeling was conducted using the density functional theory (DFT) method to investigate the adsorption performance of the catalyst and the effect of the composition and structure of the material on the kinetics of the catalytic process [50]. Theoretical calculations provided additional insight into the role of the pyridine ring in facilitating the photocatalytic 2e^−^ two-step ORR performance of COFs in the production of H_2_O_2_. The benefits of these catalysts in photocatalytic H_2_O_2_ production are determined by the interaction between OOH* intermediates and the active sites on the catalyst surface (Figure 8a,b). Figure 8c demonstrates that the energy barrier of the OOH* intermediate in TTP-COF was lowered by 0.898 eV compared to TBP-COF. This indicates that modifying the pyridine ring of the COFs can enhance the binding strength of OOH* intermediates to the active sites [51], leading to highly effective photocatalytic activity for the production of H_2_O_2_.

## 4. Conclusions

In summary, two triazine-based COFs (TBP-COF and TTP-COF) were designed and synthesized with a focus on achieving efficient photocatalytic production of H_2_O_2_ from H_2_O and O_2_. TTP-COF demonstrated excellent photocatalytic performance because of its strong light absorption, nitrogen-rich triazine and pyridine rings, and an appropriate band structure for photocatalytic H_2_O_2_ generation. The TTP-COF catalyst achieved an H_2_O_2_ production of 4244 μmol h^−1^ g^−1^ without requiring additional cocatalysts or sacrificial agents. In addition, TTP-COF demonstrated excellent cyclic stability, retaining its structural and chemical characteristics effectively after photocatalytic cycling tests. Pyridine significantly altered the band structure of TTP-COF and expanded its visible light absorption, making it more appropriate for the 2e^−^ two-step ORR process. This increased the overall photogeneration of H_2_O_2_. The DFT calculations of TBP-COF and TTP-COF demonstrate that including a pyridine ring is advantageous for enhancing photocatalytic performance. This work demonstrates that the degree of conjugation and nitrogen content of COFs can be enhanced to achieve optimal overall photocatalytic H_2_O_2_ generation. This study also provides valuable insights into the design and development of COF-based catalysts for the photosynthesis of chemical fuel.

## Data Availability

The data presented in this study are available on request from the corresponding author.

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
