# Peer review of "Nitrogen-Rich Triazine-Based Covalent Organic Frameworks as Efficient Visible Light Photocatalysts for Hydrogen Peroxide Production"

_nanomaterials, 2024, doi:10.3390/nano14070643_

Round 1

Reviewer 1 Report

Comments and Suggestions for Authors

This manuscript was well-organized and seemed to be interesting in the fields of photochemistry and COF. Overall, I recommend the publication of this article, but the following issues should be considered by the authors.

i) I am wondering how the authors calculated the AQY values for the entire wavelength region. Figure 6b & 6c show the AQY plots for TBP and TTP. In my view, those plots were very similar to the absorption spectra. 

ii) Regarding the TRPL results, Figure 7i shows the very fast PL decays in TTP and TBP with the decay constant less than 5 ns. However, such result could not support the author statements (page 12, 386 lines); "the lifetime of TTP-COF (20.02 ns) is lower than that of TBP-COF (23.45 ns)". The author should carefully check the electronic dynamics based on the TRPL results.

iii) The results from the DFT calculation looks reasonable and can support the experimental observation in the microscopic view. The formation of OOH* intermediate in TTP-COF can hugely reduce the energy barrier for the photocatalytic production compared to in TBP-COF. I think that this result is of importance in this study, thus it should be properly highlighted in the sections of Abstract or Conclusions for the potential readers. In addition, the recent studies reported the similar observations for OER or ORR process. I recommend the following studies will be properly cited in this result and will be used for the additional discussion.

- Vinogradov et. al., ACS Omega, 2022, 7, 7066-7073, DOI: 10.1021/acsomega.1c06768

- Jung et. al., J. Mat. Chem. A, 2024, DOI: 10.1039/D3TA07803E

Author Response

Dear Nanomaterials Referees,

We carefully went through those valuable comments and suggestions and made point by point response to reviewer’s comments which we hope meet with approval. In our revised manuscript, we highlighted changes we made in yellow in the revised manuscript to make it easier for referees to follow.

Reviewer 2 Report

Comments and Suggestions for Authors

The manuscript titled " Nitrogen-rich Triazine-based Covalent Organic Frameworks as Efficient Visible Light Photocatalysts for Hydrogen Peroxide Production" presents an interesting study on the synthesis and characterization of triazine-based amorphous COF materials for photocatalytic hydrogen peroxide production. While the research topic is significant and the experimental data are promising, there are several areas that require major revision before considering publication. Please consider the following comments for revision:

11. Authors should explain more about the material design and superiority of this work addressing the drawbacks of previous work.

22. Porosity of the COF materials are very important. Authors must check the BET surface areas of both the COF materials presented.

33. Please, include the experimental conditions for the stability experiment based on time, temperature, concentration etc.

44. Chemical stability of the photocatalyst is another point to consider. As the COFs are amorphous, no strong peak was observed in XRD. The reported the FT-IR study is not explained properly. More discussion about peak position is needed. In addition, for chemical stability only FT-IR analysis is not sufficient. Surface area determination before and after treatment of different solvents will be helpful to understand the stability. Particularly, the effect of HCl and NaOH is interesting for such Schiff-base based COFs. Authors may consider the following article:

i.                     Chem. Sci., 2022,13, 9655-9667

55. Authors mentioned “Elemental analysis (EA) of TBP-COF (Table 1) and TTP-COF (Table 2) confirmed 282 that the C, H, and N content of these COFs aligned well with the theoretical values.” How was the theoretical value calculated? Proper discussion is needed.

66. SEM and TEM images after catalysis along with elemental analysis should be provided to understand any changes during catalysis.

77. XPS spectra of TTP-COF after catalysis will be helpful to understand the fate of pyridine moiety after catalysis. Please, consider additional XPS analysis after catalysis.

88. Throughout the manuscript, ensure the full names of all chemicals and solvents are clearly stated upon first mention.

Comments on the Quality of English Language

A thorough revision is necessary for the English in this manuscript. There are several areas where clarity and coherence could be improved. Please review and refine the language for better readability and understanding.

Author Response

Dear Nanomaterials Referee

We carefully went through those valuable comments and suggestions and made point by point response to reviewer’s comments which we hope meet with approval. In our revised manuscript, we highlighted changes we made in yellow in the revised manuscript to make it easier for referees to follow.

Round 2

Reviewer 2 Report

Comments and Suggestions for Authors

I recommend accepting this manuscript in its present form.